# Genome-Wide Identification of the Dof Gene Family and Functional Analysis of PeSCAP1 in Regulating Guard Cell Maturation in *Populus euphratica*

**DOI:** 10.3390/ijms26083798

**Published:** 2025-04-17

**Authors:** Yongqiang Chen, Yang Yuan, Mingyu Jia, Huiyun Yang, Peipei Jiao, Huimin Guo

**Affiliations:** 1Key Laboratory of Crop Genetic Improvement, Hubei Hongshan Laboratory, Huazhong Agricultural University, Wuhan 430070, China; chenyq2017@163.com (Y.C.); yangyuan9109@163.com (Y.Y.); ly1393464544@163.com (H.Y.); 2State Key Laboratory Incubation Base for Conservation and Utilization of Bio-Resource in Tarim Basin, College of Life Science, Tarim University, Alar 843300, China; jiamingyu0717@163.com

**Keywords:** Dof, *Populus euphratica*, drought stress, *PeSCAP1*, guard cell maturation

## Abstract

DNA-binding with one finger (Dof) transcription factors plays critical roles in regulating plant growth and development, as well as modulating responses to biotic and abiotic stresses. While the biological characteristics of the Dof family have been explored across various species, their functions in *Populus euphratica* remain largely uncharacterized. In this study, we identified 43 *PeDof* family genes through a genome-wide approach, revealing a total of 10 conserved motifs across all family members. Predictions of cis-acting elements indicated that *Dof* genes are involved in light signaling, hormone signaling, and stress responses. Phylogenetic analysis classified the 43 *Dof* genes of *P. euphratica* into six distinct groups, with genes within the same group exhibiting relatively conserved structures. Expression pattern analyses demonstrated significant regulation of *PeDof* genes by drought stress, with their expression also being influenced by environmental conditions during seed germination. Furthermore, we identified the *Dof* gene *PeSCAP1*, which plays a conserved role in regulating guard cell maturation, underscoring the importance of stomatal morphology and function in leaf water retention. This study enhances our understanding of the role of *Dofs* in abiotic stress responses and provides valuable insights into their function in *Populus euphratica*.

## 1. Introduction

Transcription factors are essential components of gene regulation, modulating transcription levels of target genes by binding to specific sequences in their promoter regions. These factors intricately regulate the growth and development of plants [1]. Therefore, identifying and functionally characterizing transcription factors is crucial for comprehending and harnessing their regulatory roles. Transcription factors can be classified into various families based on the structural characteristics of their DNA-binding domains, among which the Dof (DNA binding with One Finger) gene family is notably important [2].

Dof proteins are a classical type of transcription factor belonging to the zinc finger superfamily, first identified in maize [3]. This class is unique to plants, as Dof proteins have not been found in other eukaryotic organisms such as yeast, Drosophila, Caenorhabditis elegans, or humans [4,5]. Typically comprising 200 to 400 amino acids, Dof proteins contain a conserved Dof domain within a 52-amino-acid zinc finger region at their N-terminus, while the transcriptional regulatory domain resides at the C-terminus [6]. Unlike other zinc finger proteins, Dof transcription factors possess a single Cys2/Cys2 zinc finger, which specifically recognizes the core sequence 5′-(T/A)/AAAG-3′ in the promoter regions of target genes. Compared to other transcription factor families, the recognition motifs of Dof proteins are relatively short [5], resulting in numerous potential Dof-binding sites across various gene promoter regions. The plant genome contains a limited number of Dof transcription factors, with previous studies identifying 36 and 30 *Dof* genes in *Arabidopsis* and rice, respectively [6,7,8]. Furthermore, 26, 36, 35, 22, and 24 *Dof* genes have been reported in birch [9], watermelon [10], foxtail millet [11], spinach [12], and rose [13], respectively. Nevertheless, ongoing research continues to uncover new members of the Dof gene family, with numerous functions yet to be elucidated.

Dof transcription factors are involved in various aspects of plant growth and development, including root growth, hypocotyl elongation, plant morphogenesis, leaf development, and floral organ formation [14,15,16,17,18]. For instance, CDF4 (Cyclin Dof Factor 4) in *Arabidopsis thaliana* promotes the differentiation of root column stem cells [17]. The Dof transcription factor COG1 (COGGWWEEL1) regulates brassinosteroid (BR) biosynthesis, thereby facilitating hypocotyl elongation through binding to and activating the promoters of *PIF4* and *PIF5* [18]. Recent studies have demonstrated that *Arabidopsis* CDF2 and PIF4 interact to regulate the downstream target gene *YUCCA8*, thus promoting hypocotyl elongation [16]. Furthermore, SCAP1 (STOMATAL CARPENTER 1), another Dof transcription factor, plays a crucial role in guard cell differentiation and stomatal maturation in *Arabidopsis*. The absence of *SCAP1* leads to stomata that likely cannot effectively regulate aperture, as evidenced by the scap1 mutant’s insensitivity to fluctuations in CO_2_ concentration and light intensity [14].

Water is essential for the survival of plants, as insufficient water can limit their growth. To mitigate the impacts of drought, plants have evolved strategies to minimize water loss, maintain cellular hydration, and endure drought conditions. During drought stresses, plants actively maintain physiological water balance by increasing root water uptake, reducing water loss through stomatal closure, and adapting to osmotic stress [19]. Stomata, small epidermal pores surrounded by guard cells, facilitate gas exchange and water regulation. The optimal morphology, size, and density of stomata significantly influence water use efficiency in plants, and manipulating stomatal movement or density can enhance drought resistance [19,20,21,22,23,24,25,26,27,28]. For instance, Papanatsiou et al. introduced a blue light-responsive ion channel into stomata, enhancing the dynamics of stomatal opening and closing in response to light [20]. Additionally, overexpression of the secretory peptides *EPF1* (*Epidermal Patterning Factor 1*) or *EPF2* has been shown to improve drought resistance by regulating stomatal density across various species [24,25,26].

*Populus euphratica*, predominantly located in the arid northwest of China and other desertified regions, is a dominant tree species in desert ecosystems. Its robustness allows it to thrive in arid and semi-arid environments, rendering it an ideal model for studying abiotic stress in woody plants due to its significant economic and ecological value [29,30,31]. While the biological characteristics of the Dof gene family have been explored across several species, research focusing on *Populus euphratica* remains limited. In this study, we identified all *Dof* genes in *Populus euphratica* and predicted their physicochemical properties. We also conducted analyses of gene structures, conserved motifs, cis-acting elements, multi-species collinearity, and phylogenetic relationships. To assess the involvement of PeDof in regulating drought tolerance, we evaluated expression changes in Dof genes under simulated drought conditions. Furthermore, we found that *PeSCAP1* conservatively regulates stomatal maturation, suggesting its role in maintaining water supply to plant leaves through the preservation of guard cell morphology and function.

## 2. Results

### 2.1. Identification and Prediction of the Physicochemical Properties of PeDof Genes

To identify Dof proteins in *Populus euphratica*, we employed the consensus amino acid sequence of the DNA-binding domain of Dof proteins that have been previously annotated in *Arabidopsis thaliana* [6] and conducted a BLASTP 2.12.0 search against the *P. euphratica* genome using TBtools (version 2.125) [32]. This analysis revealed a total of 43 PeDof transcription factor genes (Table 1). The presence of these proteins’ typical binding domain, defined by a 52-residue single Cys2/Cys2 zinc finger structure (Dof domain), was confirmed through HMMER (https://www.ebi.ac.uk/Tools/hmmer/search/phmmer (accessed on 2 October 2024)) (Appendix A). To further characterize these proteins, we utilized the ExPASy online tool (https://web.expasy.org/protparam (accessed on 4 October 2024)) to predict their physicochemical properties (Table 1). These results indicated that the protein lengths varied from 159 amino acids (PeuTF04G00502.1) to 509 amino acids (PeuTF17G00710.1), with molecular weights ranging from 17 kDa (PeuTF04G00502.1) to 55 kDa (PeuTF17G00710.1). The isoelectric point (pI) values of PeDof proteins ranged from 4.61 (PeuTF05G01648.1) to 9.49 (PeuTF05G01128.1). The instability index, an estimate of protein stability in vitro, varied from 36.75 to 68.67, indicating considerable variability in stability among these proteins. Furthermore, the Grand Average of Hydropathicity (GRAVY) for all proteins was negative, suggesting that these Dof proteins are hydrophilic.

### 2.2. Gene Structures and Conserved Motifs of PeDofs

We constructed a phylogenetic tree to investigate the evolutionary relationships among the 43 PeDofs. The MEME software was utilized to predict conserved motifs within these Dof proteins (Figure 1A and Appendix A). This analysis identified 10 conserved motifs, sequentially labeled from Motif 1 to Motif 10. The number of conserved motifs among *P. euphratica* Dof proteins ranged from one to six, with the majority of proteins exhibiting just one conserved motif. Motif 1 is the Dof domain, which is present in all PeDof proteins, indicating its high conservation. Other motifs were found to be distributed among closely related genes. Notably, motifs 2, 3, 7, and 10 are exclusively present in four specific genes (*PeuTF04G01143.1*, *PeuTF17G00710.1*, *PeuTF08G00503.1*, and *PeuTF10G00899.1*). An analysis of gene structures (Figure 1B) revealed that the number of exons in *PeDof* genes varied from one to three, with the predominant structure comprising one exon and no introns, followed by structures with two exons and one intron.

### 2.3. Prediction of Cis-Acting Elements in PeDofs

Given the critical role of cis-acting elements in gene regulation, we analyzed the promoter region of *PeDof* genes. This analysis revealed numerous light-responsive elements and hormone-responsive elements (specifically for jasmonic acid [JA] and abscisic acid [ABA]) (Figure 2). The identified elements included G-box, GT1 motif, Box4, TGACG, and ABRE. Additionally, nearly half of the promoter regions were found to contain MYB binding sites (MBS), which are associated with drought inducibility, suggesting a potential link between these PeDofs and the response to drought stress. These findings implied that PeDofs may play roles in hormone regulation and responses to abiotic stress.

### 2.4. Collinearity Analysis of Dofs in Multi-Species

To further explore the evolutionary dynamics of *Dof* genes in Populus, we conducted a collinearity analysis between *P. euphratica* and four other species (*A. thaliana*, *P. pruinosa*, *P. deltoides*, and *P. trichocarpa*). We identified 39, 42, and 51 *Dof* genes in *P. pruinosa*, *P. trichocarpa*, and *P. deltoides*, respectively (Figure 3A). The collinearity analysis revealed a total of 107, 116, 114, and 52 collinear Dof gene pairs between *P. euphratica* and *P. pruinosa*, *P. trichocarpa*, *P. deltoides*, and *A. thaliana*, respectively. These results suggested that collinearity among *Dof* genes is more conserved within Populus species than that observed between *P. euphratica* and *A. thaliana*.

Further analysis of Dof gene collinearity within the *P. euphratica* genome revealed extensive collinearity, implying functional similarities among most paired genes (Figure 3B). A total of 39 *PeDof* gene pairs were identified on different chromosomes, suggesting that segmental duplications have taken place in these regions, which may contribute to the expansion of the PeDof family. The distribution of all 43 *PeDofs* spanned 18 chromosomes, excluding chromosome 18 (Figure 4). Notably, chromosomes 04, 05, and 11 exhibited the highest numbers of *PeDofs*, containing 5, 4, and 5 genes, respectively. Chromosomes 02, 07, 12, and 15 each harbored three *PeDofs*, while chromosomes 01, 06, 08, 10, and 14 contained two *PeDofs* each. Additionally, single *Dof* genes were also identified on chromosomes 09, 13, 16, 17, and 19.

### 2.5. Phylogenetic Tree of PeDofs

To investigate the evolutionary relationships among *PeDof* members, we performed a phylogenetic analysis incorporating protein sequences from 36 *A. thaliana* Dof family members, 39 *P. pruinosa* Dofs, and 43 *P. euphratica* Dofs (Figure 5). The results indicated an overrepresentation of members from Dof groups II, IV, and VII, while groups III and VI were underrepresented across these genomes. Notably, both the *P. euphratica* and *P. pruinosa* genomes were found to lack Dof group I, which may reflect a loss during poplar evolutionary speciation. Following the classification scheme established for the Dof family in *Arabidopsis* [6], all 82 Dof proteins in *P. euphratica* and *P. pruinosa* were categorized into six distinct groups. The classification of PeDof proteins into distinct groups suggests that different subfamilies may have unique functional roles based on conserved amino acid motifs. Group IV contained the largest number of Dofs (26; ~32%), followed by groups II (15; ~18%), VII (15; ~18%), V (10; ~12%), VI (8; ~10%), and III (8; ~10%). No Dofs from either PeDof or PpDof were found in group I. Compared to *A. thaliana*, the Dof proteins from *P. euphratica* and *P. pruinosa* exhibited greater homology with one another, whereas *Arabidopsis* Dof proteins clustered into separate branches on the phylogenetic tree. Furthermore, we noted that almost all PeDof proteins have a very close homology to PpDof proteins within the same branch, except for one protein, PeuTF03G00284.1 in group IV, which also has no collinear protein pairs within *P. euphratica* (Figure 3B), indicating that this protein may have a specific function.

### 2.6. Expression Patterns of PeDofs in Roots and Leaves Under Drought Treatment

Given *Populus euphratica*’s reputation for its high drought resistance, we aimed to determine whether *PeDof* genes are induced under drought conditions. We analyzed RNA-seq data regarding gene expression in response to drought stress [33]. To simulate drought stress, 25% polyethylene glycol-6000 (PEG6000) was applied to the roots or leaves, with samples collected for RNA extraction at 4 and 12 h post-treatment. Expression profiling revealed that most *PeDof* genes in leaves are downregulated under drought conditions, with only six genes (*PeuTF02G00480.1*, *PeuTF05G01114.1*, *PeuTF11G00355.1*, *PeuTF12G00528.1*, *PeuTF14G01584.1*, and *PeuTF15G00470.1*) exhibiting upregulation at 4 h (Figure 6). Among them, we noticed a sharp decrease in *PeuTF10G00899.1* (from 20.02 to 0.87) and a sharp increase in *PeuTF12G00528.1* (from 6.31 to 47.58) at 4 h (Figure 6). The significant changes in the expression levels of these two genes may play an important role in drought stress. In roots, over half of the *PeDof* genes were downregulated, with only five genes (*PeuTF02G00480.1*, *PeuTF05G01114.1*, *PeuTF11G00355.1*, *PeuTF14G01584.1,* and *PeuTF15G00759.1*) showing upregulation at 4 h (Figure 6). Moreover, *PeuTF05G01114.1* and *PeuTF08G00173.1* showed the fastest response at 4 h (Figure 6). To evaluate whether these *Dof* genes might work together under drought stress, we grouped genes with similar expression changes. Our analysis revealed multiple expression patterns, including genes like *PeuTF11G00355.1* and *PeuTF02G01562.1*, which had high expression in both roots and leaves and similar response patterns; genes such as *PeuTF19G00358.1* and *PeuTF16G00617.1* showing very low expression in both tissues; and genes including *PeuTF05G01114.1* and *PeuTF02G01099.1*, exhibiting lower expression in leaves but higher expression in roots (Figure 6). These represented some of the observed coordinated expression patterns, suggesting potential functional collaboration among *PeDof* genes during drought stress. Together, these results indicated significant involvement of *PeDof* genes in drought response.

### 2.7. PeDof Genes Response to Drought or ABA Treatment During Seed Germination

Seed germination is highly sensitive to environmental conditions. To investigate the expression patterns of *PeDofs* under drought stress and ABA treatment, *P. euphratica* seeds were treated with 15% PEG 6000 (to simulate drought) or 100 µM ABA for transcriptome sequencing as previously documented [34]. RNA-seq analysis revealed that among the 43 *Dof* genes (Figure 7), 4 genes (*PeuTF02G00480*, *PeuTF05G01648*, *PeuTF12G00528*, and *PeuTF14G01584*) were upregulated and 18 genes downregulated following PEG treatment. In addition, after ABA treatment, only 1 gene (*PeuTF02G01099*) was upregulated, while 24 genes were downregulated. These findings indicated that PeDofs may be implicated in responses to drought stress or ABA during seed germination. Drought and ABA treatment inhibited seed germination, correlating with the reduced expression levels of most *PeDof* genes (Figure 7), implying an important role for *PeDof* genes in this process. Additionally, we classified genes with similar expression patterns after PEG or ABA treatment and found that some genes (*PeuTF02G01099.1*, *PeuTF03G00284.1*, etc.) showed a decrease in expression levels after PEG treatment, while they increased or remained unchanged after ABA treatment. Some genes (*PeuTF01G02355.1*, *PeuTF15G00759.1*, etc.) were downregulated by PEG or ABA treatment. Some specific genes, such as *PeuTF05G01648*, *PeuTF11G00428*, and *PeuTF16G00617*, exhibited low expression levels (Figure 7), suggesting they may not be expressed during seed development. Collectively, these results supported a potential regulatory role for *PeDof* genes in seed germination.

Promoter analysis of *PeDof* genes revealed that 15 gene promoter regions contain MBS sequences (Figure 2), indicating responsiveness to changes in arid environments. Under PEG conditions, seven genes were downregulated, while two genes were upregulated. Under ABA conditions, seven genes were downregulated, with only one showing upregulation. These observations implied that *PeDof* genes harboring MBS cis-elements may significantly contribute to responses to drought or ABA.

### 2.8. PeSCAP1 Is Responsible for Maintaining the Integrity of the Morphology and Function of Guard Cells

Integrating the promoter and transcriptional analyses of *PeDof* genes in response to drought stress (Figure 2, Figure 6 and Figure 7), we hypothesized that Dof proteins are closely associated with the drought tolerance exhibited by *P. euphratica*. The *Arabidopsis Dof* gene *SCAP1* has been demonstrated to directly regulate guard cell maturation; the *scap1* mutant exhibited irregular guard cell morphology and heightened drought sensitivity [14]. This highlights the critical role of stomatal morphology in plant water retention and drought resistance. Therefore, we were prompted to investigate whether *PeSCAP1* functions conservatively in regulating stomatal development in *P. euphratica*. The gene *PeuTF07G00620*, which shares significant homology with *AtSCAP1*, has been designated *PeSCAP1* (Figure 5 and Appendix A). To validate the function of *PeSCAP1*, we assessed its subcellular localization, revealing that the YFP signal of *PeSCAP1* co-localizes with the nuclear localization signal of H2B-CFP (Figure 8A), indicating that *PeSCAP1* is located in the nucleus.

To assess the conservation of *PeSCAP1’s* function, we introduced *35S:: PeSCAP1* into the *atscap1-3* mutant and generated the transgenic plants *35S::PeSCAP1/atscap1-3* (*PeSCAP1*-*com*) (Appendix A). We subsequently analyzed the phenotypes of *atscap1-3* and *PeSCAP1*-*com* lines; the *atscap1* mutant exhibited aberrant guard cell morphology (Figure 8B), whereas the proportion of morphologically abnormal guard cells in *PeSCAP1*-*com* plants was reduced (Figure 8C). This observation indicated that *PeSCAP1* partially rescues the defective phenotype associated with the *AtSCAP1* mutation. Consequently, this evidence suggested that *PeSCAP1* is relatively conserved compared to *AtSCAP1*, facilitating guard cell maturation and maintaining morphological integrity. As previously reported, the loss of *AtSCAP1* led to rapid leaf dehydration due to the impaired ability of stomata to close effectively [14], highlighting the critical role of intact guard cells in leaf water retention. Therefore, we assumed that *PeSCAP1* is crucial for water retention and drought tolerance in *P. euphratica*. To verify this hypothesis, we conducted water loss experiments on the leaves of Col-0, *atscap1-3*, and *PeSCAP1*-*com* lines (Figure 8D). The results showed that the *atscap1-3* leaves were very prone to dehydration and wilting, while the supplemented plants (*PeSCAP1*-*com*) exhibited a phenotype similar to the wild-type (Col-0). Overall, these results suggested that the integrity of guard cell morphology and function is crucial for water retention.

## 3. Discussion

*Populus euphratica* is a perennial tree species prevalent in the arid regions of northwestern China, where drought stress poses a significant challenge to its survival. Understanding the drought resistance mechanisms of this species is crucial for the sustainable and economic utilization of land resources in arid areas. The *Dof* genes, as a plant-specific transcription factor family, play an important role in the growth and development of all plants. Increasing evidence underscores the critical role of Dof proteins in a variety of biological processes, including plant tissue differentiation, seed development, metabolic regulation, and responses to environmental stressors [2,35]. Although Dof proteins have been identified in several species, including rice, Chinese cabbage, apple, and tea tree, their presence in *P. euphratica* had not previously been documented. This study presented the first genome-wide identification and characterization of the *Dof* gene family in *P. euphratica*. We systematically analyzed the gene structures, conserved motifs, cis-regulatory elements, phylogenetic relationships, and expression profiles of these transcription factors (Figure 1, Figure 2, Figure 3, Figure 4, Figure 5, Figure 6 and Figure 7). Furthermore, we investigated the functional conservation of the Dof transcription factor *PeSCAP1* in stomatal development (Figure 8).

Compared to other transcription factor families, such as MYB and WRKY, the Dof transcription factor family has relatively fewer members. Our analysis identified 43 PeDof proteins in *P. euphratica*, with amino acid lengths ranging from 159 to 509 residues (Table 1). The proteins exhibited considerable variability in isoelectric points and stability, indicating diverse functional roles of Dof proteins in plant biology. The examination of gene structures and conserved motifs revealed ten conserved motifs within Dof proteins, with motif 1 being universally present across all proteins. The remaining motifs tended to be distributed among proteins with similar evolutionary relationships (Figure 1). Previous studies have indicated that different Dof gene subgroups can fulfill distinct functional roles, attributed to variations in amino acid sequences outside the conserved Dof domain [6,36]. Our phylogenetic analysis, which included Dof proteins from *Arabidopsis*, *P. euphratica*, and *P. pruinosa*, classified PeDof and PpDof proteins into six distinct subgroups, with subgroup members sharing analogous gene structures (Figure 5). These observations further underscore the structural and functional diversity of Dof proteins. These 43 *PeDof* genes were distributed on 18 chromosomes (Figure 4), and intraspecific collinearity analysis showed that 39 gene pairs are collinear (Figure 3), suggesting that the involved chromatin regions may have undergone segmental duplication, which may contribute to the expansion of the Dof family.

What are the functional roles of the Dof transcription factor family in *P. euphratica*? Prior studies have highlighted the multifaceted roles of Dof proteins in regulating plant growth, development, and abiotic stress responses [2,35,37]. For instance, the tomato gene *SlDof22* is involved in ascorbic acid accumulation and enhances salt tolerance [38]; *Arabidopsis Dof5.8* is implicated in the salt signaling pathway [39]; and several *MaDof* genes in banana exhibit downregulation under salt and drought stress [40]. Notably, Dof family genes also contribute significantly to drought resistance in woody plants. For example, overexpression of *Dof54* in apples has been shown to increase plant survival rates under short-term drought conditions [41]. In the current study, we identified numerous cis-acting elements linked to various potential functions within these promoter regions of Dof genes in *P. euphratica*, particularly highlighting light-responsive and hormone-responsive elements (Figure 2). Furthermore, nearly half of the promoter regions contained MYB binding sites associated with drought inducibility (Figure 2), suggesting a potential link between these *PeDofs* and drought stress responses. Expression pattern analyses revealed that *PeDof* genes are responsive to drought stress (Figure 6). Among them, genes that can respond rapidly to drought changes, such as *PeuTF10G00899.1* and *PeuTF12G00528.1*, might be used as marker genes for drought stress. Additionally, during seed germination, *PeDof* gene expression was induced by drought or ABA treatment (Figure 7), further supporting their involvement in drought stress regulation.

Plants respond to drought stress through the regulation of stomatal movement and development [19,20,21,22,23,24,25,26,27,28]. To explore the function of *Dof* genes in *P. euphratica*, we studied the Dof transcription factor *PeSCAP1*, which regulates stomatal maturation and maintains stomatal morphological integrity (Figure 8). Previous studies in *Arabidopsis* have shown that *AtSCAP1* controls stomatal maturation by regulating the expression of genes related to cell wall composition and stomatal movement [14]. We conducted preliminary verification and found that *PeSCAP1* may function through a similar regulatory pathway (Appendix A). Impaired guard cell function can result in inadequate stomata responses to environmental changes; for example, if stomata fail to close properly during drought conditions, plants may suffer from rapid water loss and subsequent wilting. In this article, the water loss experiment of detached leaves further confirmed the above hypothesis (Figure 8D) and proved that *PeSCAP1* plays a critical role in preserving leaf hydration through the maintenance of guard cell morphology and function. These findings suggested that manipulating stomatal development or *SCAP1* homologous genes in crops could potentially enhance water use efficiency and drought tolerance. The evolutionary conservation of Dof family genes implies that *Dof* genes in other species, including rice, maize, and wheat, may play broad roles in abiotic stress responses.

## 4. Materials and Methods

### 4.1. Genome-Wide Identification and Physiochemical Predictions of the P. euphratica Dof Gene Family

The conserved Dof domain amino acid sequences from the *Arabidopsis thaliana* Dof gene family were employed to identify Dof genes within the *Populus euphratica* genome [6]. Identification was based on genomic data specific to *P. euphratica* [42]. Dof genes from four additional poplar species were identified using genomic data from *P. pruinosa* (NCBI BioProject accession number PRJNA863418), *P. deltoides* (WV94_445) [43], and *P. trichocarpa* (V3.1) [44]. The identified *PeDof* genes were further validated using HMMER (https://www.ebi.ac.uk/Tools/hmmer/search/phmmer (accessed on 2 October 2024)). Additionally, the physiochemical properties of the *P. euphratica* Dof genes were predicted utilizing the ExPASy online tool (http://web.expasy.org/protparam/ (accessed on 4 October 2024)), assessing parameters such as amino acid count, molecular weight, theoretical isoelectric point, instability index, aliphatic index, and grand average of hydropathicity (GRAVY).

### 4.2. Gene Structure and Conserved Motifs Analysis

Conserved motif prediction for PeDof proteins was conducted using the MEME Suite (version 5.5.7) (http://meme-suite.org (accessed on 2 October 2024)). The maximum number of motifs was set to 10, with motif width restricted between 10 and 150 amino acids, while other parameters remained at default settings. The resulting gene structures and conserved motifs of all *P. euphratica* Dof family members were visualized and analyzed using TBtools software (version 2.125).

### 4.3. Analysis of Cis-Acting Elements in the Promoter Regions of PeDof Genes

Cis-acting elements within the promoter regions of the *PeDof* family were analyzed by utilizing the DNA sequences located 2000 bp upstream of the initiation codon (ATG) for all 43 *PeDof* genes. The PlantCare website (https://bioinformatics.psb.ugent.be/webtools/plantcare/html (accessed on 2 October 2024)) was employed to predict the cis-elements present in each promoter region. Subsequent visualization of these predictions was executed using TBtools (version 2.125).

### 4.4. Multispecies Collinearity and Chromosomal Localization of PeDofs

BLAST-P alignment was performed to identify orthologous pairs between *P. euphratica* and four other species (*P. pruinosa*, *P. trichocarpa*, *P. deltoides*, and *A. thaliana*). Collinearity analyses included both intraspecific and interspecific BLAST comparisons. Duplicated gene pairs were identified from the interspecific analysis, treating *P. euphratica* separately from the other species. The resulting duplication events were visualized as collinearity relationships using covariance circles in TBtools (version 2.125). Chromosome length data (Fasta Stats), *PeDof* gene IDs, and positional information (GFF3 gene position parse) from the *P. euphratica* genome file were utilized in TBtools (version 2.125) for chromosome position visualization.

### 4.5. Phylogenetic Tree Analysis of PeDofs

Phylogenetic trees for Dof proteins from *Arabidopsis*, *P. pruinosa*, and *P. euphratica* were constructed using the MUSCLE method in MEGA-X (version 11.0.13) [45]. The evolutionary history was inferred via the neighbor-joining method [46]. The bootstrap consensus tree, resulting from 1000 replicates, represented the evolutionary relationships among the taxa analyzed [47]. Branches corresponding to partitions represented in less than 50% of bootstrap replicates were collapsed. Evolutionary distances were calculated using the JTT matrix-based method [48], expressed as the number of amino acid substitutions per site, with rate variation modeled by a gamma distribution (shape parameter = 1). This analysis included 118 amino acid sequences, with all ambiguous positions removed via pairwise deletion, resulting in a final data set of 734 positions.

### 4.6. Expression Analysis of PeDofs Under Drought Stress and ABA Treatment

The RNA-seq data for drought stress and ABA treatment were obtained from a previous study [34]. In the experiment, *Populus euphratica* seeds were subjected to treatment with 15% (*v*/*v*) PEG6000 or 100 µmol/L ABA in an incubator maintained at 25 °C to 30 °C under a 16h/8h light/dark photoperiod. RNA was subsequently extracted from the seeds for RNA-seq experiments. Transcriptome data are accessible from the National Genomics Data Center (NGDC, https://ngdc.cncb.ac.cn/ (accessed on 7 October 2024)), under project accession number PRJCA007522. The expression levels of *PeDofs* in both control and treatment groups were extracted, analyzed, and visualized using TBtools (version 2.125).

### 4.7. Subcellular Localization of PeSCAP1

To ascertain the subcellular localization of PeSCAP1, the full-length coding sequence (CDS) was amplified and cloned into the *pGreen0179-35S-MCS-YFP* vector to generate the *Pro35S:PeSCAP1-YFP* construct [49]. All primers used are listed in Appendix A. This construct, along with pCambia1300-P19, was transformed into *Agrobacterium* GV3101 and subsequently infiltrated into the leaf epidermis of *Nicotiana benthamiana*. After a 3-day incubation at 21 °C, imaging of epidermal cells was performed using a confocal microscope (TCS-SP8; Leica), excited with a 514 nm laser, with emitted signals detected between 524 and 574 nm.

### 4.8. Plant Materials and Growth Conditions

In this study, *Arabidopsis thaliana* Col-0 served as the genetic background. The *scap1-3* mutant, containing an 8 bp deletion in the *SCAP1* gene coding region (Appendix A), was derived from the T-DNA insertion line SALK_111683. Notably, SALK_111683 represents the *at5g50850 scap1-3* double mutant. To generate *35S::PeSCAP1/atscap1-3* complementation lines (*PeSCAP1-com*), the CDS of *PeSCAP1* was amplified and cloned into the *pGreen0179-35S-MCS-YFP* vector. The resulting *35S::PeSCAP1* construct was then transformed into *atscap1-3* mutant plants to produce transgenic complementation lines. Plants were cultivated on ½ Murashige and Skoog (MS) medium supplemented with 1% (*w*/*v*) sucrose at 21 °C under a 16 h light/8 h dark photoperiod with a light intensity of 80 μmol m^−2^ s^−1^.

### 4.9. Stomatal Phenotype Quantification and Water Loss Assay

To calculate the proportion of different types of stomata, we analyzed the eighth rosette leaf from four-week-old seedlings. The types of stomata were classified into two distinct categories: (1) normal (wild-type morphology) and (2) abnormal (collapsed phenotype, as observed in *atscap1* mutants). For each genotype, ≥100 stomata were counted to determine the percentage distribution of these morphological classes.

For the detached leaf water loss assay, similarly sized leaves from plants of comparable age were selected (n ≥ 3 leaves per genotype). Leaves were excised and photographed immediately (0 h) and after 5 h of ambient exposure. The degree of wilting of leaves reflected the water retention capacity of different plant leaves.

## 5. Conclusions

In conclusion, we identified 43 Dof family transcription factors in *Populus euphratica.* Analysis of their gene structures and conserved motifs revealed diversity within this protein family, while phylogenetic analysis further demonstrated the genetic variations among these Dof genes. Cis-acting element analysis indicated that these transcription factors may be involved in hormone regulation and abiotic stress responses. Notably, expression pattern analyses showed significant regulation of both root and leaf Dof genes under drought stress, along with substantial changes during seed germination. The Dof gene *PeSCAP1* was found to conservatively regulate guard cell maturation, highlighting the importance of stomatal integrity for maintaining leaf water retention. Our findings demonstrated the potential utility of *PeDof* genes for improving drought resistance in crops or woody plants and provided valuable genetic resources for molecular breeding of Populus species.

## Figures and Tables

**Figure 1 ijms-26-03798-f001:**
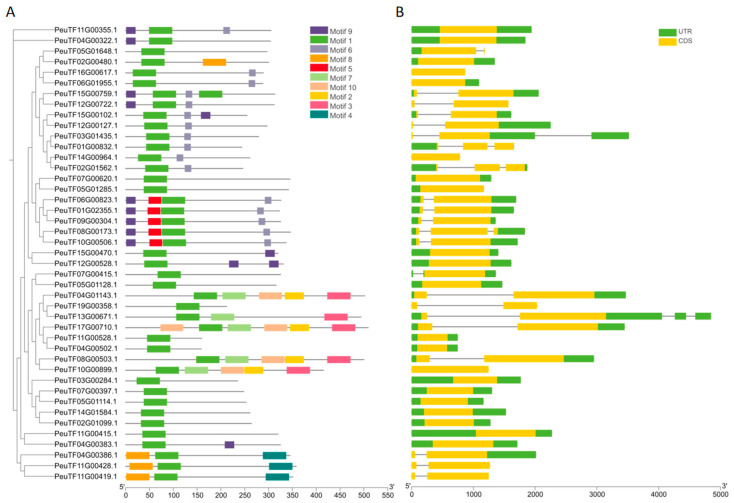
Conserved motifs and gene structures of PeDofs. (**A**) Arrangement of conserved motifs in *PeDof* genes, with ten distinct motifs labeled in various colors. Conserved motifs were identified by uploading amino acid sequences to the MEME online tool (http://meme-suite.org (accessed on 2 October 2024)) and displayed using TBtools (version 2.125). The amino acid information of conserved motifs was provided in Appendix A. (**B**) Gene structures of *PeDof* genes, the structures were drawn on genomic lengths by using *Populus euphratica* GFF3 files and visualized using TBtools (version 2.125). The green boxes denote 5′ and 3′ untranslated regions, yellow boxes represent coding sequences, and black lines mark the introns. The scale can determine the length of the introns and exons at the bottom.

**Figure 2 ijms-26-03798-f002:**
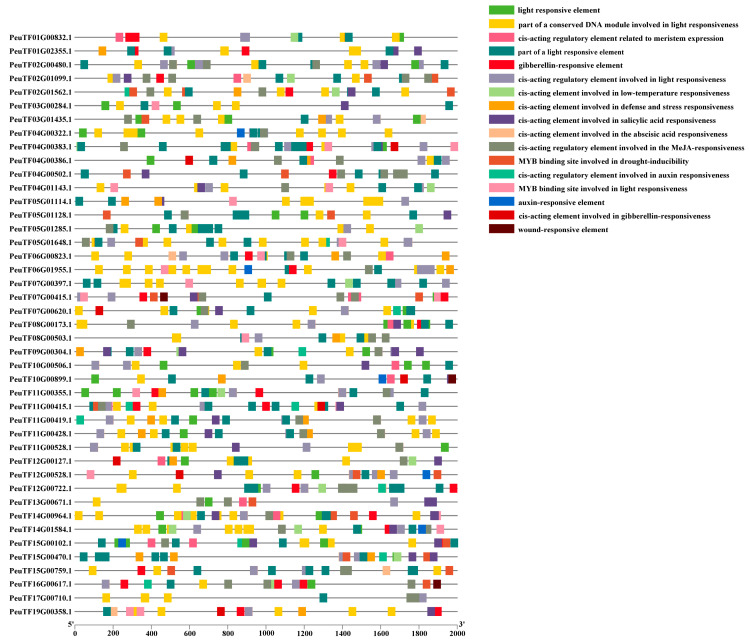
Cis-elements in the promoter region of *PeDof* genes. Various color bars represent different cis-elements.

**Figure 3 ijms-26-03798-f003:**
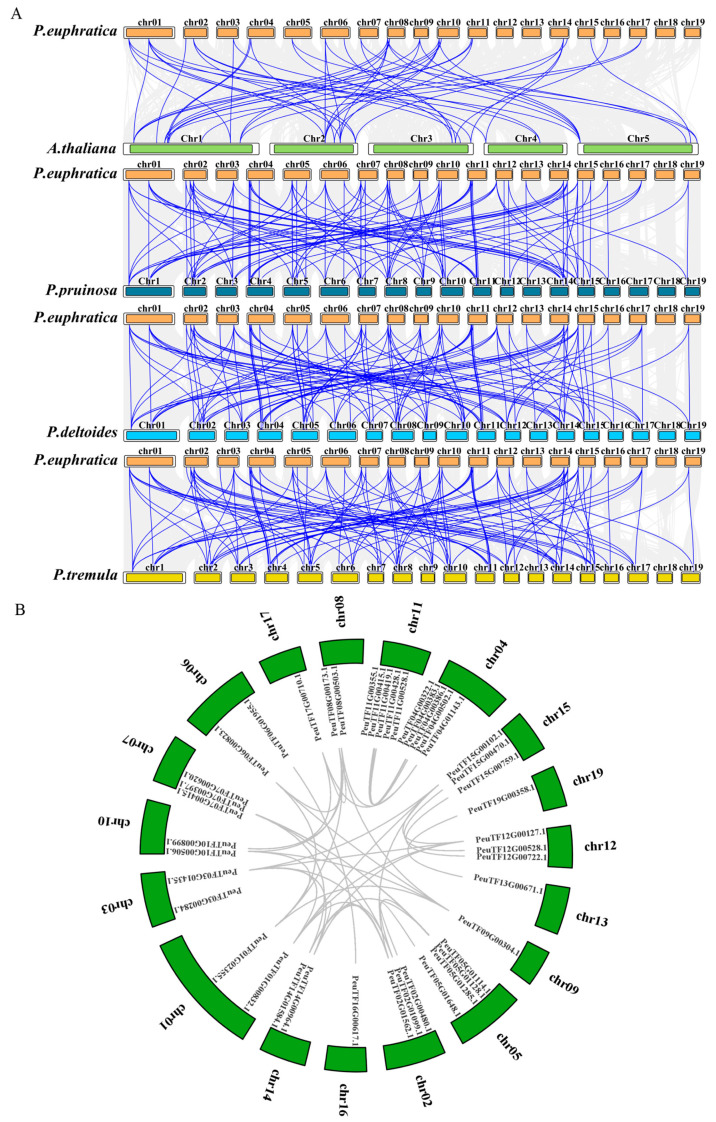
Collinearity analysis of *Dof* genes across multiple species. (**A**) Collinearity analysis of *Dof* genes between *P. euphratica* and four other species (*A. thaliana*, *P. pruinosa*, *P. trichocarpa*, and *P. deltoides*). Gray lines indicate collinear blocks, while blue lines highlight collinear Dof gene pairs. (**B**) Intraspecific collinearity analysis of *PeDof* genes, with gray lines representing collinear *Dof* gene pairs.

**Figure 4 ijms-26-03798-f004:**
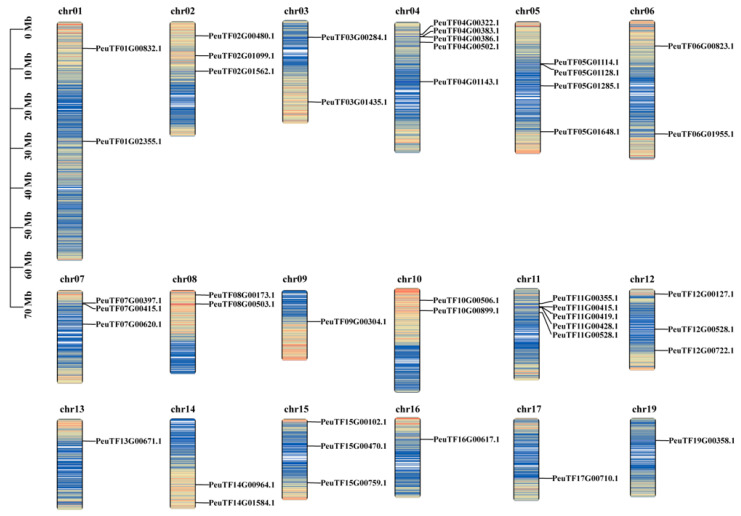
Chromosomal distribution of *PeDofs* in the *P. euphratica* genome. Blue bars indicate low gene density, while red bars indicate high gene density across chromosomes.

**Figure 5 ijms-26-03798-f005:**
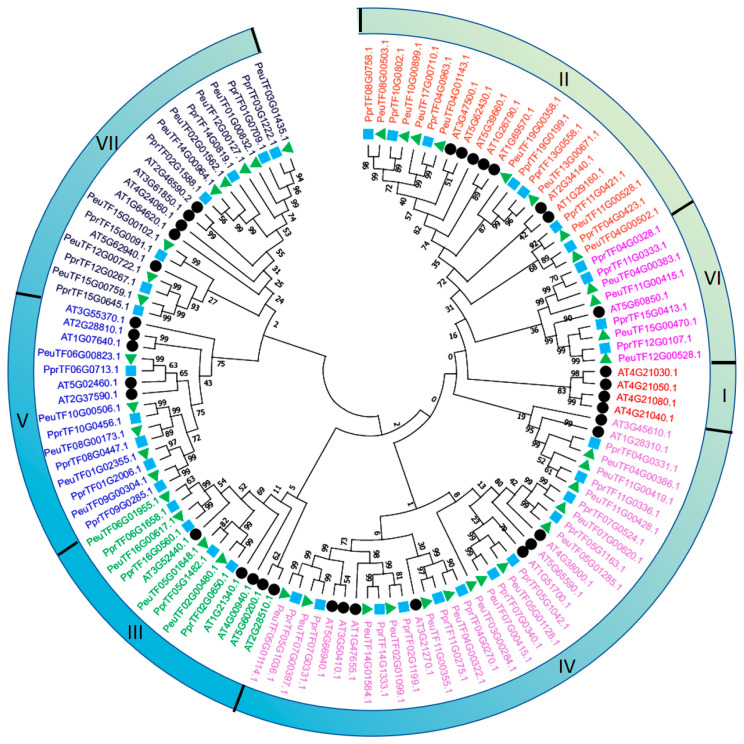
Phylogenetic tree of Dof proteins among *P. euphratica*, *P. pruinosa*, and *A. thaliana*. The green triangle, blue square, and black triangle represented the Dof proteins of *P. euphratica*, *P. pruinosa*, and *A. thaliana*, respectively. The phylogenetic tree was constructed using the neighbor-joining (NJ) method in MEGA-X (version 11.0.13) (1000 bootstrap replicates). This analysis included 118 amino acid sequences, with all ambiguous positions removed via pairwise deletion, resulting in a final data set of 734 positions. Dof proteins from different subgroups were marked with distinct colors, revealing seven groups designated as I–VII.

**Figure 6 ijms-26-03798-f006:**
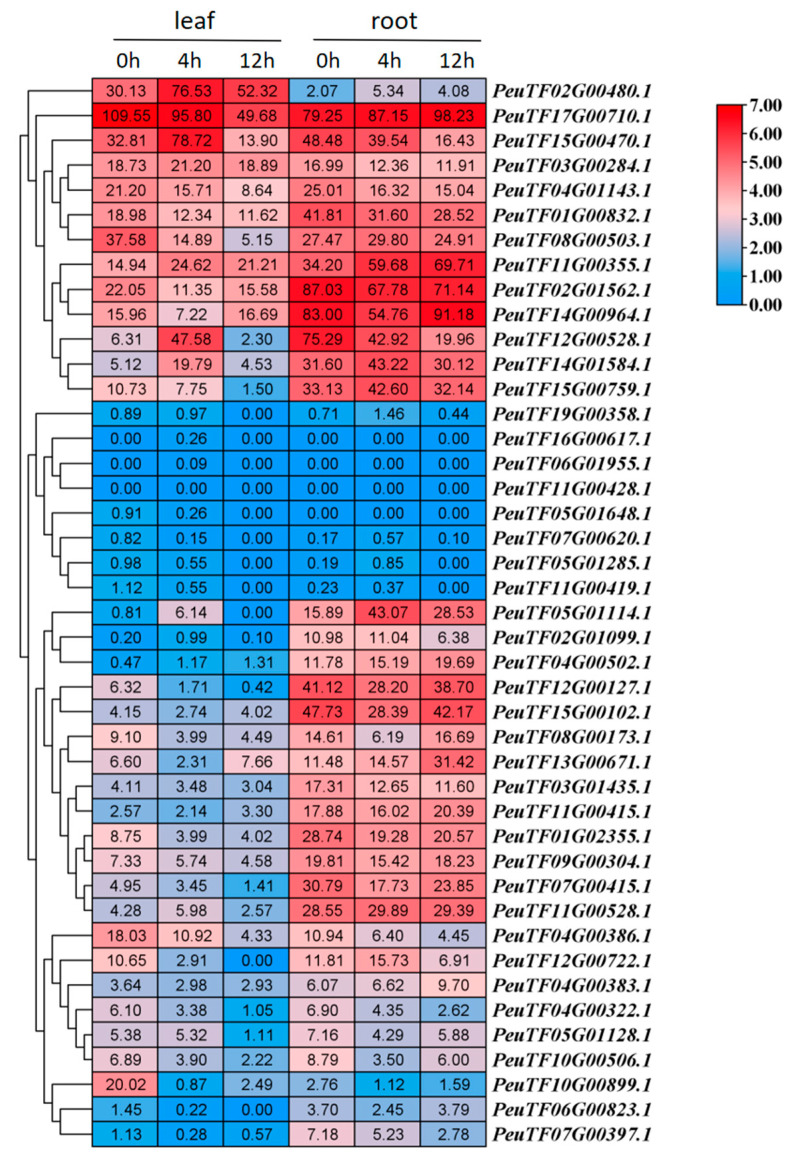
Expression patterns of *PeDof* genes in response to drought stress (PEG treatment) at 0 h, 4 h, and 12 h. Colors in the heatmap represent gene transcript levels, as indicated by the key bar to the right of the figure.

**Figure 7 ijms-26-03798-f007:**
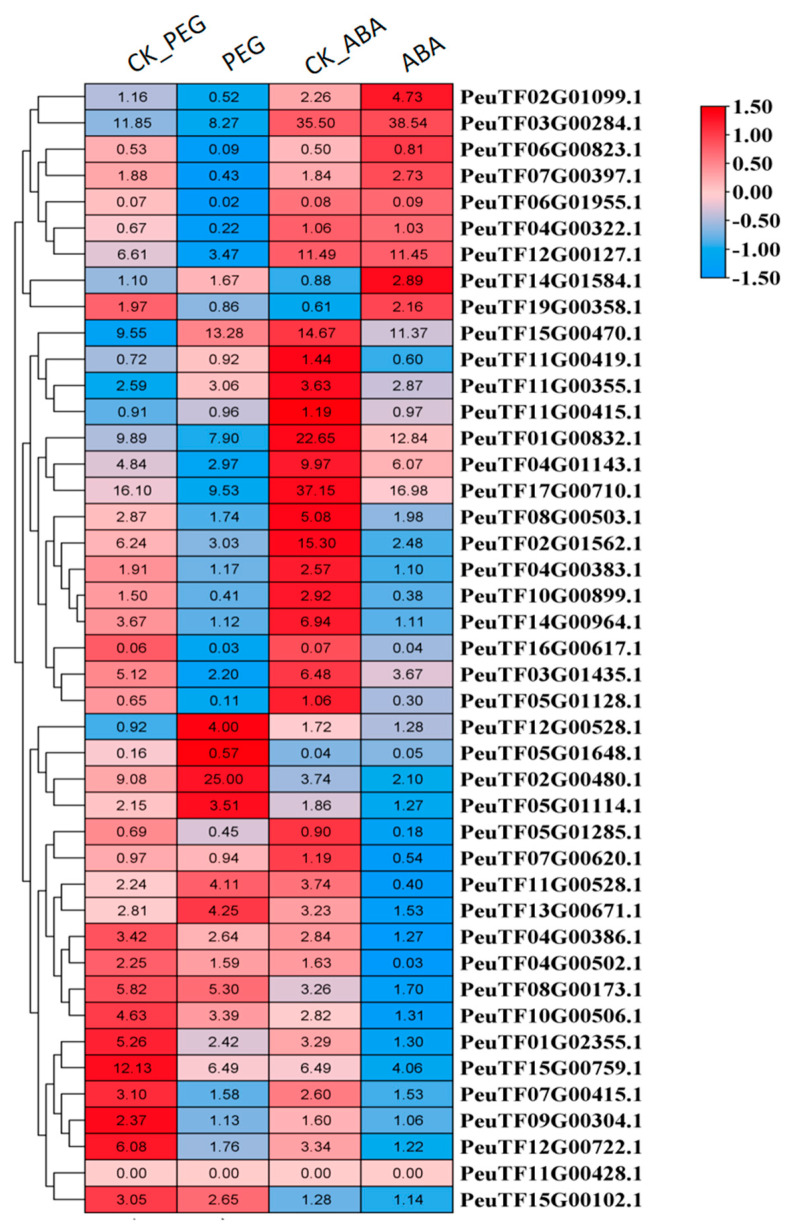
Expression patterns of *PeDof* genes under drought stress and ABA treatment during seed germination. Heatmap colors denote gene transcript levels, as detailed in the key bar to the right of the figure.

**Figure 8 ijms-26-03798-f008:**
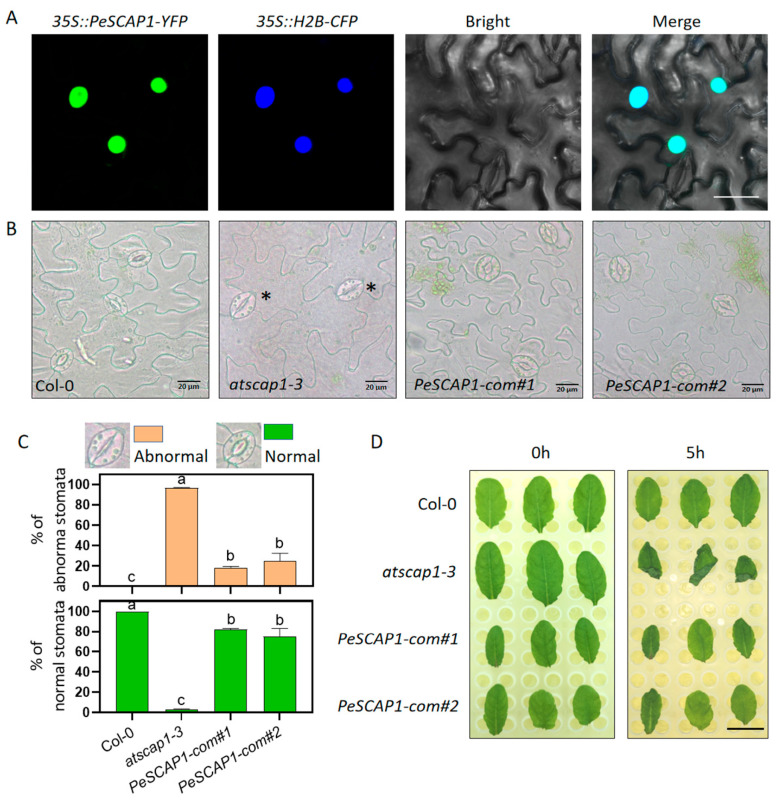
The transcription factor *PeSCAP1* in stomatal development regulation. (**A**) Subcellular localization of PeSCAP1 in tobacco epidermis. The SCAP1-YFP signal (green) co-localized with the nuclear localization signal of H2B-CFP (blue), as shown in the merged fluorescence image (cyan). Scale bar = 30 µm. (**B**) Morphological analysis of abaxial epidermis from Col-0 (wild-type), *atscap1*, and *PeSCAP1*-complemented lines (*PeSCAP1*-*com*) at 4 weeks, highlighting abnormal stomata indicated by black stars. Scale bar = 20 µm. (**C**) Statistics on the percentage of different types of stomata displayed on the epidermis of Col-0, *atscap1-3*, and *PeSCAP1*-*com* plants. One-way ANOVA with Tukey’s test was conducted for statistical analyses. Approximately 100 stomata were counted and classified into normal and abnormal types across six seedlings per genotype. Different letters indicated statistically significant differences among genotypes at a significance level of *p* < 0.05. (**D**) Water loss in the detached leaves of Col-0, *atscap1-3*, and *PeSCAP1*-*com* lines for 5 h. The degree of wilting of leaves reflected the water retention capacity of different plant leaves. Scale bar = 15 mm.

**Table 1 ijms-26-03798-t001:** Characteristics of the putative 43 *PeDof* genes.

Gene ID	Number of Amino Acids	Molecular Weight	Theoretical pI	Instability Index	Aliphatic Index	Grand Average of Hydropathicity (GRAVY)
PeuTF01G00832.1	244	27,021.21	8.17	52.46	63.11	−0.671
PeuTF01G02355.1	323	34,585.48	9.48	63.19	59.16	−0.583
PeuTF02G00480.1	300	34,034.88	4.77	53.38	63.40	−0.675
PeuTF02G01099.1	264	27565.24	6.30	45.47	46.97	−0.541
PeuTF02G01562.1	246	27,015.15	9.28	46.80	53.17	−0.748
PeuTF03G00284.1	235	25,200.50	8.79	46.44	43.62	−0.930
PeuTF03G01435.1	279	30,697.23	8.63	52.31	62.58	−0.641
PeuTF04G00322.1	304	33,730.41	8.73	53.13	61.88	−0.791
PeuTF04G00383.1	325	35,635.15	9.3	61.05	52.80	−0.916
PeuTF04G00386.1	345	38,108.04	8.42	53.79	54.23	−0.757
PeuTF04G00502.1	159	17,653.88	8.99	49.20	49.62	−0.826
PeuTF04G01143.1	502	54,941.55	5.85	54.11	50.54	−0.894
PeuTF05G01114.1	253	26,039.95	8.44	41.24	56.72	−0.378
PeuTF05G01128.1	316	33,705.48	9.49	45.06	63.96	−0.516
PeuTF05G01285.1	342	37,143.24	8.92	44.88	59.68	−0.608
PeuTF05G01648.1	297	33,543.19	4.61	43.52	61.08	−0.695
PeuTF06G00823.1	326	34,715.73	9.10	54.35	50.95	−0.542
PeuTF06G01955.1	288	31,953.02	5.98	62.88	50.80	−0.826
PeuTF07G00397.1	248	25,508.40	8.57	41.51	60.24	−0.344
PeuTF07G00415.1	325	34,483.32	8.96	49.11	63.97	−0.452
PeuTF07G00620.1	345	37,195.00	8.28	44.45	56.35	−0.599
PeuTF08G00173.1	346	36,955.26	9.18	51.97	63.99	−0.545
PeuTF08G00503.1	500	54,216.45	6.51	56.44	48.60	−0.833
PeuTF09G00304.1	325	34,474.38	9.35	62.58	51.32	−0.524
PeuTF10G00506.1	337	35,634.61	9.30	50.81	58.72	−0.569
PeuTF10G00899.1	415	45,414.71	9.33	60.22	50.72	−0.782
PeuTF11G00355.1	305	33,930.65	7.57	58.18	61.05	−0.775
PeuTF11G00415.1	320	35,125.52	9.26	68.67	50.91	−0.912
PeuTF11G00419.1	351	38,445.75	8.73	49.97	58.06	−0.646
PeuTF11G00428.1	358	39,314.79	8.57	50.56	59.11	−0.642
PeuTF11G00528.1	160	17,804.13	9.28	56.11	54.19	−0.778
PeuTF12G00127.1	297	32,814.02	7.14	48.17	46.40	−0.844
PeuTF12G00528.1	331	36,602.88	6.30	49.99	53.90	−0.911
PeuTF12G00722.1	312	34,251.20	6.64	43.32	57.47	−0.628
PeuTF13G00671.1	494	53,843.54	6.57	46.71	60.61	−0.580
PeuTF14G00964.1	261	28,561.58	9.14	58.13	49.35	−0.808
PeuTF14G01584.1	261	27,480.08	5.84	51.41	46.36	−0.572
PeuTF15G00102.1	255	28,187.27	8.24	50.45	47.49	−0.770
PeuTF15G00470.1	320	35,176.36	7.70	49.66	54.25	−0.838
PeuTF15G00759.1	313	34,601.65	6.19	49.33	57.57	−0.650
PeuTF16G00617.1	289	32,307.57	5.77	57.43	50.93	−0.764
PeuTF17G00710.1	509	55,583.03	5.49	51.25	49.86	−0.913
PeuTF19G00358.1	212	23,388.24	7.60	36.75	57.08	−0.787

## Data Availability

No large datasets were created in this study.

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
