# Peer review of "Genome-Wide Identification of the Dof Gene Family and Functional Analysis of PeSCAP1 in Regulating Guard Cell Maturation in Populus euphratica"

_ijms, 2025, doi:10.3390/ijms26083798_

Round 1
Reviewer 1 Report
Comments and Suggestions for Authors
The manuscript “Genome-wide Identification of the Dof Gene Family and Mechanistic Insights into PeSCAP1 Regulation of Guard Cell Maturation in Populus euphratica” explores the identification and functional analysis of the Dof (DNA-binding with One Finger) gene family in Populus euphratica, focusing particularly on the role of PeSCAP1 in regulating guard cell maturation and drought stress responses. The study presents a comprehensive analysis of 43 PeDof genes, including their gene structures, conserved motifs, phylogenetic relationships, and expression patterns under drought stress conditions. The key findings highlight the significant role of PeSCAP1 in stomatal development and drought resistance.
- The study contains gene expression data, although several results need further analysis. While drought stress-related gene expression is discussed, more information is needed on how the identified genes interact and contribute to drought resistance.
- Without explanation, conserved motif and evolutionary tree representations are hard to understand. Figure captions and supporting information would improve clarity.
- While the manuscript provides some mechanistic insights, there is little discussion on how these findings could be translated into broader agricultural practices or their implications for other species.
- In section 4.6, the authors discuss the expression patterns (using transcriptomics) of PeDof genes under drought stress and ABA treatment. However, it is unclear whether this data originates from this study or was taken from previous research. The manuscript would benefit from a clearer explanation of whether the expression data is from the authors’ own experiments or from other studies. If the data is indeed part of this study, the authors should provide more detailed results and methods, including specific experimental conditions (e.g., concentration of PEG and ABA, duration of treatment, sample collection time points, and the methodology used for RNA-seq analysis). Including these details would allow for better assessment and reproducibility of the findings.
- The expression data and phenotypic observations (e.g., in the guard cell assays) would benefit from statistical tests to assess the significance of the observed differences. A more rigorous statistical approach would strengthen the claims regarding the role of PeSCAP1 and other PeDof genes.
- The study involves genetic manipulation of PeSCAP1, but it does not address potential off-target effects or compensatory mechanisms that may influence the observed phenotypes in the transgenic lines. A more thorough discussion of possible confounding factors would make the results more robust.
- The manuscript identifies PeSCAP1 as important for stomatal development, but it does not fully elucidate the molecular mechanisms through which PeSCAP1 regulates guard cell maturation. A deeper analysis of its molecular targets, possible interactions with other transcription factors, or signaling pathways could provide a more comprehensive understanding.
- The phylogenetic analysis focuses on the classification of PeDof genes in Populus euphratica but does not discuss how these genes have evolved in the context of other species, particularly related species in the Salicaceae family.
- The manuscript contains some awkward phrasing and minor grammatical errors that could affect readability. A careful proofreading would improve the overall flow and clarity of the paper.
The manuscript contains some awkward phrasing and minor grammatical errors that could affect readability. A careful proofreading would improve the overall flow and clarity of the paper.
Reviewer 2 Report
Comments and Suggestions for Authors
Dear Authors,
I have reviewed the manuscript and have the following observations.
The topic of the manuscript is the study of single-finger DNA-binding (Dof) transcription factors in stress responses.
The manuscript is novel in that 43 PeDof family genes have been identified in Populus euphratica using a genome-wide approach, revealing a total of 10 conserved motifs in all family members - thus, they may contribute greatly to the study of stress responses in this species.
The manuscript represents a significant contribution to the field, as expression pattern analyses have shown that PeDof genes are significantly regulated by drought stress and their expression is also influenced by environmental conditions during seed germination.
My comments on the manuscript are as follows:
I suggest rewriting the Discussion section. This section is short and compares our own results with few results. I suggest moving from the general to the specific and comparing these with your own results. Much deeper context is needed here.
The Conclusions chapter should draw general conclusions for the future - preferably in a separate, stand-alone chapter and in a deeper context.
Round 2
Reviewer 1 Report
Comments and Suggestions for Authors
Comments on the Quality of English Language
The authors have improved the manuscript according to the suggestions with possible explanations of the data used and the methodology adapted.
Author Response
We sincerely appreciate the reviewer's insightful comments, which have significantly improved the scientific rigor, clarity, and overall quality of our manuscript.
Reviewer 2 Report
Comments and Suggestions for Authors
I recommand it for publication.
Author Response

(The authors gave the same response as above.)
